# Visual Adaptations in Predatory and Scavenging Diurnal Raptors

**Simon Potier**

Department of Biology, Lund University, Sölvegatan 35, S-22362 Lund, Sweden; sim.potier@gmail.com

**Abstract:** Ecological diversity among diurnal birds of prey, or raptors, is highlighted regarding their sensory abilities. While raptors are believed to forage primarily using sight, the sensory demands of scavengers and predators differ, as reflected in their visual systems. Here, I have reviewed the visual specialisations of predatory and scavenging diurnal raptors, focusing on (1) the anatomy of the eye and (2) the use of vision in foraging. Predators have larger eyes than scavengers relative to their body mass, potentially highlighting the higher importance of vision in these species. Scavengers possess one centrally positioned fovea that allows for the detection of carrion at a distance. In addition to the central fovea, predators have a second, temporally positioned fovea that views the frontal visual field, possibly for prey capture. Spatial resolution does not differ between predators and scavengers. In contrast, the organisation of the visual fields reflects important divergences, with enhanced binocularity in predators opposed to an enlarged field of view in scavengers. Predators also have a larger blind spot above the head. The diversity of visual system specializations according to the foraging ecology displayed by these birds suggests a complex interplay between visual anatomy and ecology, often unrelatedly of phylogeny.

**Keywords:** birds of prey; foraging; predators; scavengers; vision

## 1. Introduction

Understanding how animals extract information from their surrounding environment is essential to appreciate how they complete their daily tasks such as finding food, avoiding predators, attracting mates, and navigating their environment [1]. Among the sensory organs found in the animal kingdom, eyes can provide instantaneous information about the environment like no other [2]. While birds use a wide range of cues (ranging from the Earth's magnetic field to sounds, odours, visual cues, and so on), vision appears to be a very important sensory modality in these animals, especially in the context of foraging [3]. In particular, birds of prey, or "raptors" (as defined below), are considered to forage primary using the visual sense ([4,5], but see [6]). Indeed, a number of species of raptors spot their prey/food at relatively high altitudes where acoustic and olfactory cues should be undetectable, and vision may be the only accessible cue available for prey/food detection in the absence of visual barriers. This apparent reliance on vision is underlined by the fact that some raptor species have the highest visual acuity (also called spatial resolution) found to date, for both achromatic ([7,8], reviewed in [5]) and chromatic [9] patterns.

While precise terminology is essential in science, defining which species belong to "raptors" (or birds of prey) has been debated for decades [10]. Derived from the Latin word "rapere", "raptor" means plunderer or ravisher. Recently, McClure et al. (2019) defined raptors as "all species within orders that evolved from a raptorial landbird lineage and in which most species maintained their raptorial lifestyle as derived from their common ancestor" [10]. This definition, based on phylogeny, ecology, and morphology, includes species in the orders of Accipitriformes (hawks, eagles, Old World vultures, kites), Cathartiformes (New World vultures), Strigiformes (owls), Falconiformes (falcons and

caracaras), and the unexpected Cariamiformes (seriemas) (see Figure 1). Because they include species that do not belong to a monophyletic clade [11] and with important morphological and ecological differences, raptors may differ significantly in their sensory abilities.

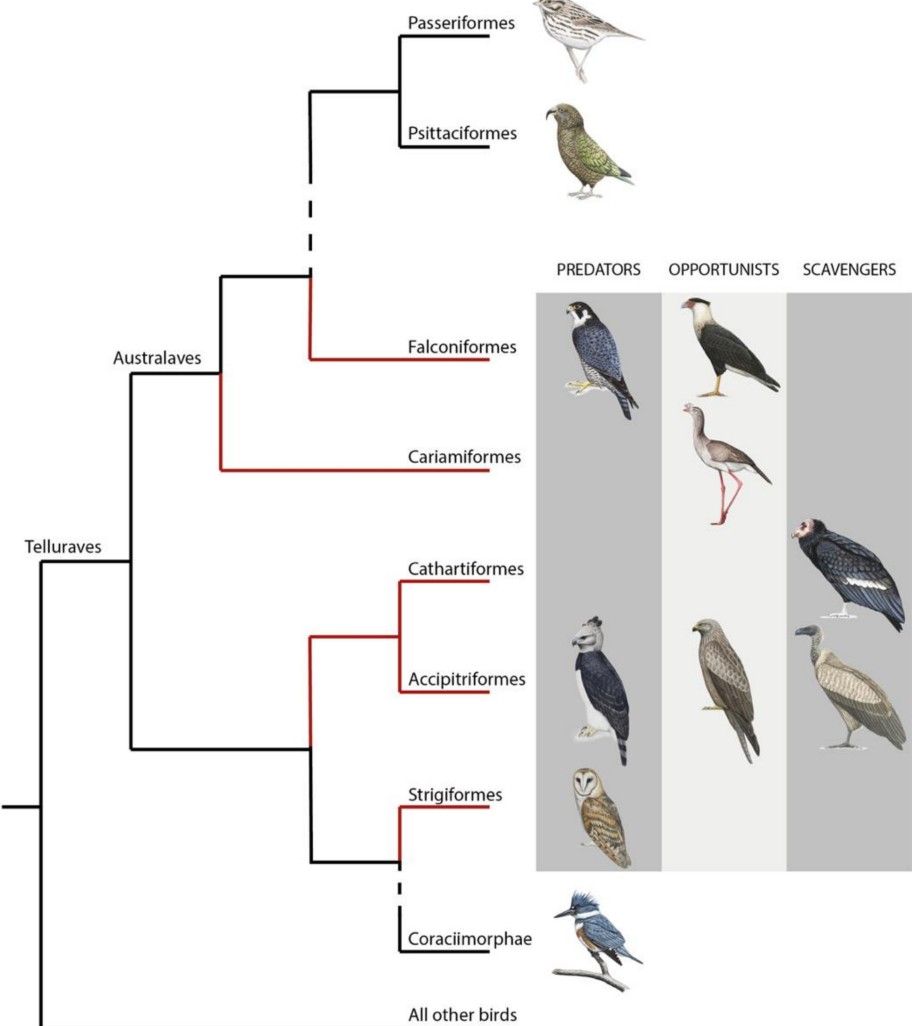

**Figure 1.** Phylogeny of core landbirds modified from [10]. The red branches encompass the order considered as raptors. For each order belonging to raptors, the presence/absence of the three foraging diet categories (predators, opportunists and scavengers) is represented by a drawing of a chosen species. "Raptor" is a paraphyletic group where species mostly share the raptorial lifestyle passed down from their single common ancestor [10]. This assumes that raptorial lifestyle has been lost twice independently with the ancestor of both Coraciimorphae and Passeriformes/Psittaciformes clades. Coraciimorphae contains six orders: Coliiformes (mousebirds), Trogoniformes (trogons), Coraciiformes (roller, kingfishers, and bee-eaters), Piciformes (woodpeckers), Leptosomiformes (cuckoo-rollers), and Bucerotiformes (hoopoes and hornbills). Drawings from Bryce W. Robinson.

A notable ecological difference among diurnal raptors is that some are active predators, whereas others are obligate scavengers. While a number of predators may scavenge (e.g., Steppe eagle *Aquila nipalensis*, Common buzzard *Buteo buteo*, and others [12]) and a small number of obligate scavengers occasionally predate (e.g., White-headed vulture *Trigonoceps occipitalis* [13]), those events are not preponderant. Consequently, scavengers and predators are likely to differ markedly in their sensory demands.

In this review, I present the differences and similarities in the visual systems of predatory and scavenging diurnal raptors, focusing exclusively on Accipitriformes, Cathartiformes, and Falconiformes. Interestingly, while the order Accipitriformes contains predators, opportunists and obligate scavengers, the order Cathartiformes only contains obligate scavenger species. Furthermore, the order Falconiformes only contains predators and some opportunist species (caracaras). Cariamiformes were not included in this review as very little (if anything) is known about their visual abilities. I also decided to not include owls (from Strigiformes) because of their nocturnal lifestyle, which implies different specialization/adaptation of the visual system [5]. Furthermore, even though some owls may scavenge on occasion [14–17], there are no obligate scavenging owl species, making the comparison between scavengers and predators impossible in these birds.

Previous reviews described the visual system of raptors [4,5], however, they did not concentrate on the differences and similarities among raptors with different foraging tactics. The aim of this review is to concentrate on visual adaptations of diurnal raptors with different lifestyle. Throughout the review, raptors were categorized as predators, scavengers, or opportunists according to Wilman et al. (2014) [18]. When species were not present in Wilman et al. (2014), foraging specialization was categorized according to Ferguson-Lees and Christie (2001) [12]. Specifically, all vultures were considered as scavengers. Caracaras and species that scavenge at least 40% of their time were considered as opportunists. Finally, species that scavenge occasionally (<20% of their time) were considered as predators. Furthermore, I also decided to highlight the lack of knowledge in the visual abilities of raptors, especially of scavengers. Emphasizing the sensory specializations of raptors with different foraging ecology and the little knowledge in scavenging species may have a crucial impact on the conservation programs of raptors.

## 2. Anatomical Specialization of the Eye

### 2.1. Predators Have Larger Eyes Than Scavengers Relative to Their Body Mass

In general, birds have big eyes in both relative (compared with body mass) and absolute terms [19], and raptors have relatively larger eyes than other birds [5,20]. Large eyes have long focal lengths and subsequently larger retinal images, and thus a potential for higher spatial resolving power in diurnal animals [2,21]. As a result, the relatively large eyes of raptors indicate the importance of the visual system for their daily life, especially because important costs are associated with increased eye size: (1) increased risk of being damaged, (2) mechanical and aerodynamic constraints [20], (3) higher metabolic and energetic costs [22], or (4) disability glare because of increased direct sunlight in the absence of adnexa (e.g., eye lids, eye brows [23]).

Interestingly, relative to their body mass, scavengers and opportunists have significantly smaller eyes than predators [24] (Figures 2A and 3). This would suggest that scavengers and opportunists invest less in vision and potentially might not need as high spatial resolution of vision as predatory species. However, from behavioural and anatomical studies, there is no evidence for lower spatial resolution in scavengers, except for Cathartiformes, whose eyes are also smaller than those of other groups (but not significantly when controlling for phylogeny [5]). Because the neural structures compete for space in the brain [25] and larger eyes may require a greater proportion of brain space dedicated to vision, scavengers and opportunists may invest more in other sensory modalities. For instance, Cathartiformes have larger olfactory bulbs than other raptors [26], and most species in which olfactory abilities have been shown are scavengers (see [6] for a review). However, olfactory bulbs appear to be freer to vary in size irrespective of other sensory structures [27] and very little is known about olfactory abilities in raptors in general [6]. More investigations are needed to understand why scavengers have smaller eyes than predators.

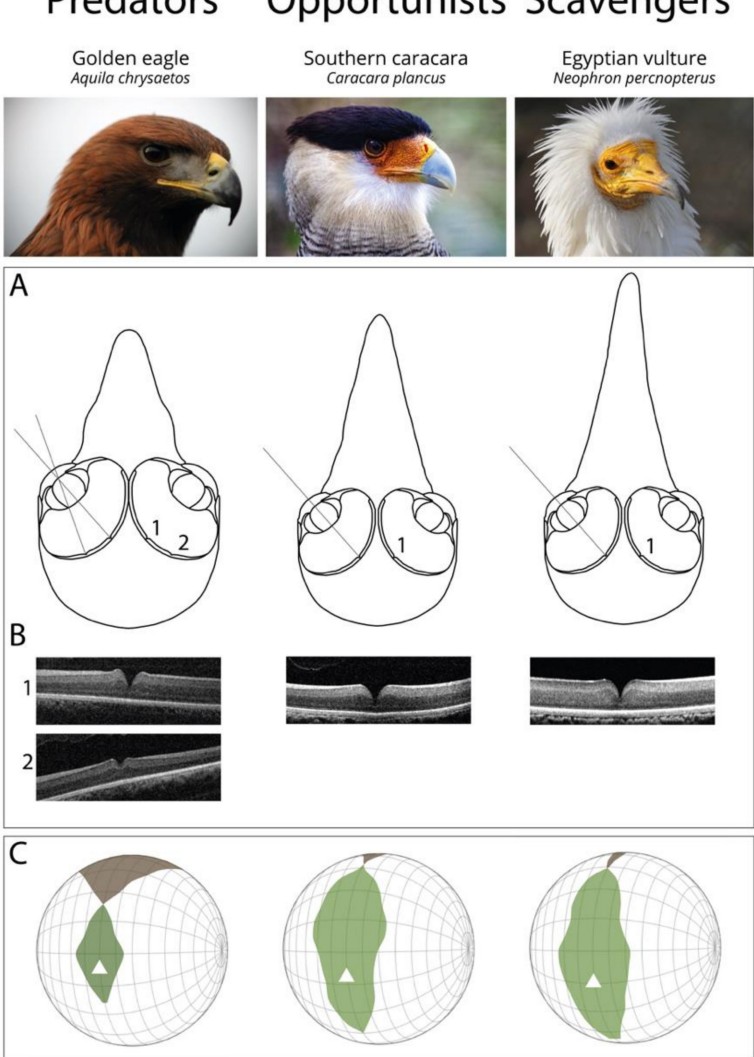

**Figure 2.** Functional differences of the visual system of raptors from different foraging tactics. (**A**) Schematic representation of frontal sections of the three chosen species (Golden eagle *Aquila chrysaetos*, Southern caracara *Caracara plancus*, and Egyptian vulture *Neoprhon percnopterus*) at the foveal plane. Fovea(s) and the centre of the pupil in each eye are on the plane. Grey lines represent the lines of sight of (1) the deep central fovea and (2) the shallow temporal fovea. Figures re-drawn from [28]. (**B**) Spectral domain optical coherence tomography (SD-OCT) images (B-scans) of the (1) central and (2) temporal fovea(s). Note that the Southern caracara and the Egyptian vulture lack temporal foveas. (**C**) Orthographic projection of retinal field boundaries of the eyes. A latitude and longitude coordinate system was used with the equator aligned vertically in the median sagittal plane (20 deg intervals in latitude and 10 deg intervals in longitude). The bird's head is at the centre of the globe. Green areas represent the binocular sector, white areas represent the monocular sectors, and brown areas represent the blind sectors. Triangles: direction of bill projection. Figures modified from [29,30]. Photography of the species was free of right thanks to @myb777_photography for the Golden eagle, @wal_172619 for the Southern caracara, and @pixel_mixer for the Egyptian vulture.

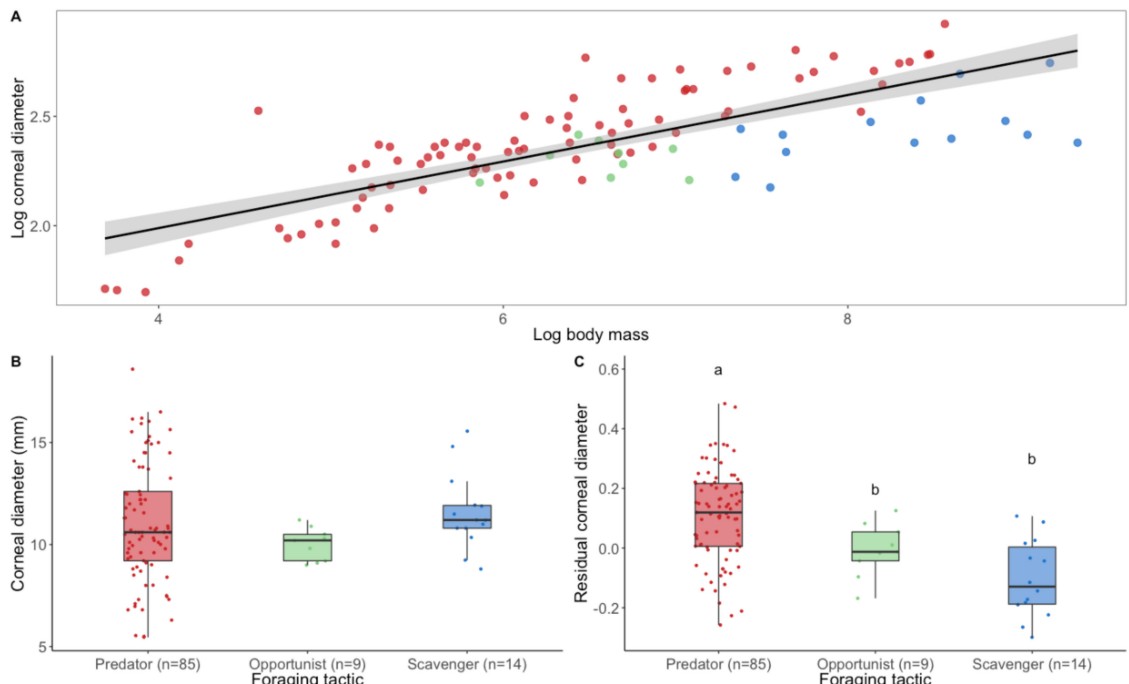

**Figure 3.** The eye size of raptors according to their foraging tactic. (**A**) Logarithmic relationship (black line) and 95% confidence level interval (grey shade) between corneal diameter (a proxy for eye size) and body mass in raptors (estimate = 0.14 ± 0.01, *t* = 9.48, *p* < 0.001). (**B**) Corneal diameter and (**C**) residual corneal diameter calculated from corneal diameter scaled to body weight in relation to foraging tactics. Differences among foraging tactic were tested using phylogenetic linear regression. The phylogenetic relationships among 130 species were estimated using a consensus tree based on 100 randomly selected trees from www.BirdTree.org [11] using Ericson tree distribution. Data were analysed on R 4.0.0 using ggplot2 [31], phylolm [32], phytools [33], caper [34], lmtest [35], ggpubr [36], and plyr [37]. Edge lengths were obtained by computing the mean edge length for each edge in the consensus tree. Model selection based on AICc and likelihood ratio test (lrtest function from lmtest package [35]) showed no differences for corneal diameter among foraging tactics (Chisq = 0.52, *p* = 0.77). By contrast, a significant difference was found for residual corneal diameter among foraging tactics (Chisq = 24.47, *p* < 0.001). Predators have significantly larger eyes compared with their body mass than opportunists (estimate = −0.11 ± 0.05, *t* = −2.20, *p* = 0.03) and scavengers (estimate = −0.21 ± 0.04, *t* = −4.90, *p* < 0.001). Scavengers and opportunists do not differ (estimate = −0.09 ± 0.06, *t* = −1.50, *p* = 0.14). Dots represent species. Different colours represent different foraging tactics (red = predator, green = opportunist, blue = scavenger). Different letters represent significant difference. Body mass were taken from [38] and corneal diameters from [27,39–41]. (Boxplots: black lines represent the median, coloured boxes represent the interquartile (IQR) range from 25th (Q1) to 75th (Q3) percentile, whiskers represent Q1 − 1.5 * IQR and Q3 + 1.5 * IQR). Note: for foraging tactics classification, please refer to Supplementary Table S1.

## 2.2. A Shared Optical System

All raptors, like all vertebrates, have camera-type eyes. Incoming light passes through the ocular media (cornea, aqueous humor, lens, and vitreous humor) and finally reaches the retina [2]. The eye can be described optically in a simple manner by the anterior focal length (which is correlated to axial length) and the pupil aperture [42]. Because pupil diameter sets the optical cut-off frequency for resolving power, the larger the pupil, the lower the diffraction limit and the higher the possible spatial resolution. Even in bright light conditions, raptors do not close their pupils [43], which again highlights the need for high spatial resolution in these species (but see [42] for optical aberrations).

The cornea and the lens function to focus the image on the retina. The accommodative power (in Diopter, i.e., the measure of the vergence of light corresponding as the reciprocal metre of the focal length) of the eye allows to maintain a clear image—or to focus on an object—as its distance varies. Therefore, a high range of accommodation is necessary for species that need to perform fine visual details at both close and long range and species that exploit different environments, such as cormorants that exploit both terrestrial and aquatic environments [44]. Birds can accommodate with the lens and the cornea [44]. Overall, across vertebrates, accommodative ability is related to lifestyle, with nocturnal species having lower accommodative power than diurnal animals [45]. The total accommodative power has been estimated in only one scavenger (Turkey vulture *Cathartes aura*: 8.5 Dioptres (D)) and six predators (Bald eagle *Haliaeetus leucocephalus*: 6.8 D; African fish eagle *Haliaeetus vocifer*: 9 D; Golden eagle *Aquila chrysaetos*: 6.7 D; American kestrel *Falco sparverius*: 16 D; Sharp-shinned hawk *Accipiter striatus*: 4 D; Red-tailed hawk *Buteo jamaicensis*: 25.8 D) in diurnal raptors [46]. It is still not clear why the Red-tailed hawk has significantly higher accommodative power than other diurnal raptors. Greater accommodation has only been measured in aquatic birds (e.g., 70–80D in waterfowls [47]). While other studies should be conducted, the accommodation capacity of the Turkey vulture falls in the range of predatory species. Glasser et al. (1997) suggested that all raptors need similar accommodative power in order to accurately position the beak for tearing at a carcass or a prey, and that this may be a stronger determinant of accommodative ability than catching prey with talons [46]. This hypothesis is supported by the similar accommodative power in seed-pecking birds (e.g., up to 17D in the chicken [48]), but the lower accommodation in owls (0–2D [46]), which swallow the whole prey and do not need accurate beak position.

## 2.3. Predators, but Not Scavengers, are Bifoveate

Raptors have the inverted retina design found in all vertebrates [2,24]. Raptors have a thicker central retina (400–500 μm thick [24]) than other birds (200–350 μm [49]). However, the retina is significantly thicker in predators [24] (Figure 2B), potentially highlighting their average higher peak retinal ganglion cell (RGC) densities [50–52]. Retinal ganglion cell density reflects the spatial resolving power of a species, and this technique has been used in Cathartiformes species [52]. We might expect predators to have higher spatial resolving powers than scavengers in view of their higher peak RGC (but see below). However, in species (such as raptors) with a fovea—an invagination in the inner retina where the photoreceptor density is the highest [53]—the RGC/cone ratio is 1:1 in the central fovea, indicating the cone density, not RGC density, limits spatial resolution [54,55].

The function of the physical structure of the fovea is still under debate [53,56], but interestingly, the number and position of fovea(s) vary in birds. All diurnal raptors studied so far possess at least one central fovea (a fovea that is centrally placed in the retina). The central fovea in raptors has been described as "convexiclivate" by Walls [57], who suggested that its steep slopes, together with the different refractive indices of the vitreous and retina, would serve as magnifying the image, and thus increase spatial resolution. Snyder and Miller (1978) suggested that this is only true for the bottom part of the foveal pit [58]. Predators and scavengers both display similar central foveal depth [24], which, according to Walls and Snyder and Miller's theories, would strongly indicate that both groups have similar magnification power. However, recent evidence suggests that fovea organisation in raptors is even more complex, as Potier et al. (2020) found that foveal shape varies significantly with age and eye size within one predatory species, the Black kite *Milvus migrans* [59].

Other authors have suggested that image magnification by the fovea is uneven [7,60] and should distort the image [61], except in the very bottom of the fovea, thus facilitating visual fixation. Interestingly, all predatory raptors studied to date (except the Broad-winged hawk *Buteo platypterus* [62]) possess a second fovea placed temporally in the retina, which is linked to frontal vision (Figure 2A,B) [24,51]. In contrast, scavengers lack this temporal fovea and, interestingly, opportunists differ in the presence or absence of the temporal fovea. Because the fovea should improve visual fixation [61], the temporal fovea has been suggested to be important for prey fixation at the

moment of capture where frontal vision (Figure 2A,B) is necessary for accurate foot positioning [63]. As scavengers do not forage on highly manoeuvrable prey, they may not need a temporal fovea. This theory is supported by the ecological convergence between active predatory raptors and some non-raptorial species that pursue highly manoeuvrable prey, which also possess a temporal fovea, such as Least terns *Sternula antillarum* [50], Sacred kingfishers *Halycon sancta*, Laughing kookaburras *Dacelo novaeguineae* [64], and Tree swallow *Tachycineta bicolor* [65].

## 3. Foraging by Sight

### 3.1. The Spatial Resolution of Vision Does Not Depend on the Foraging Mode

Since Fox's work on falcon (American kestrel *Falco sparverius*) visual acuity, raptors have been considered to have a visual acuity at least three times that of humans (160 c/deg, i.e., 160 black and 160 white vertical bars in one degree of visual angle [66]). However, a few years later, Hirsch (1982) showed that the visual acuity of the American kestrel had been considerably overestimated by Fox et al. (1976), with a correct value closer to 40 c/deg [67].

To date, the visual acuity has been estimated in eleven diurnal raptor species, including four predators: Wedge tailed eagle *Aquila audax*, Harris's hawk *Parabuteo unicinctus*, American kestrel, and Brown falcon *Falco berigora*; five scavengers: Turkey vulture, Black vulture *Coragyps atratus*, Indian vulture *Gyps indicus*, Griffon vulture *Gyps fulvus*, and Egyptian vulture *Neophron percnopterus*; and two opportunists: Black kite *Milvus migrans* and Chimango caracara *Milvago chimango*. While different methods have been used (behaviour and anatomical estimation using either cone photoreceptor or RGC spacing), visual acuity has been found to vary from 15 c/deg in the Turkey vulture [52] to 142 c/deg in the Wedge tailed eagle [7]; see [4] for a review. A detailed table can be found in [5]. Interestingly, in behavioural studies on both for scavengers and predators, visual resolution drops rapidly as light levels fall [7,8,60].

Based on these 11 species, variation in visual acuity, together with eye size, seems to reflect foraging differences, with species that forage from high altitudes (large eagles and large Old World vultures) having higher spatial resolution [7,8] compared with species that forage at low altitudes (Turkey vultures and black vultures [52]; Black kites [68]; American kestrel [67]), or species that forage from a perch (Harris's hawks [9,68]) or directly on the ground (Chimango caracara [69]). Having said this, based on the currently available data, scavengers, predators, and opportunists do not differ in their spatial resolution, despite the fact that predators have larger eyes and thicker retinas at the edge of the fovea. In mammals, it has also been found that visual acuity does not differ according to diet [70]. In a foraging context, predators, scavengers, or opportunists thus have similar capabilities to initially spot visual targets such as prey, carcasses, or conspecifics. Differences should (and do) occur after initial detection, where predators need to chase and follow their prey, a task that should be facilitated by the ability to fixate, high temporal resolution, and an enlarged binocular visual field.

### 3.2. Enlarged Field of View in Scavengers, Enlarged Binocularity in Predators

The space around an animal from which visual information can be extracted is defined by the visual field [3]. The visual field can be described by four main parameters: (1) the monocular field, the visual field of a single eye; (2) the binocular field, the area where both monocular fields overlap; (3) the cyclopean field, the total visual field produced by the combination of both monocular fields; and (4) the blind area, the space around the head from which visual information cannot be extracted [23]. In birds, visual fields vary substantially across species with different ecology [3]. For instance, while the Puna ibis *Plegadis ridwayi* (tactile forager) and the Northern bald ibis *Geronticus eremita* (visual forager) are closely related, their visual fields differ significantly, with a broader frontal binocular field in the Northern bald ibis [71]. This illustrates the trade-off between the requirement for a broader frontal binocular field for visual control of the beak/feet and the ability to gain comprehensive visual coverage for predator and conspecific detection, which has also been found in closely related ducks [72] and shorebirds [73].

This trade-off is also important in raptors. For example, species that chase highly manoeuvrable prey would be expected to possess an enlarged binocular field for visual control and accurate feet position, while scavengers would need an enlarged visual coverage for the detection of conspecifics and predators (vultures can be predated by aerial predators [74,75]). Overall, the visual fields of 18 species of diurnal raptor have been studied, including nine predators, seven scavengers, and two opportunists [4]. A recent comparative study has shown that the blind spot over the head is thinner in scavengers than in predators that hunt terrestrial prey [30] (Figure 2C). The larger blind spot above the head of predators allows better prey detection by avoiding sun dazzling. By contrast, the thinner blind spot over the head, and thus the enlarged visual coverage, of scavengers allows better conspecific detection and social foraging [76]. Predators and scavengers also differ in their binocular field shape, with a more protruding binocular field in predators [30] (Figure 2C). This is probably a result of a wider and shorter bill in species that forage on mammals [77] and enlarged optic adnexa (eyelashes and ridge above the eyes), commonly found in large-eyed species [78] to avoid sun dazzling [79].

However, some Old-World vultures (especially large species) also possess enlarged binocular fields and blind spots above the head, such as *Gyps* vultures [79] and the White-headed vulture [80], which is one of the only vulture species that have been observed to possess hunting behaviour [13]. The large blind spot above the head in *Gyps* vultures has been suggested to increase the risk of collision with wind turbines, because those vultures have a blind sector in the direction of their travel when foraging on the wing [81].

## 4. A Lack of Knowledge of Visual Abilities of Raptors, Especially of Scavengers

Little is known about the visual abilities of raptors. For example, we currently have estimates of visual acuity for less than 2% of all described raptor species (11 of the 557 raptors species [82]). This is even more true for scavengers. While visual acuity, and the organisation of the visual fields and/or the retina, has been assessed in an approximately equal number of predatory and scavenging raptor species, other visual aptitudes have been only studied in predatory species. In the following sections, I highlight the important gaps of the scavengers' visual abilities.

### 4.1. Contrast Sensitivity

Contrast sensitivity, which is a measure of how much a pattern must vary in contrast to be seen, has been estimated in only three predatory diurnal raptors: the wedge-tailed eagle (13.6 [83]), the American kestrel (30 [67]), and the Harris's hawk (12.7 [9]). While all bird species studied so far have low contrast sensitivity (from 6 to 30 [9,84]) compared with mammals (e.g., 100 in the cat [85]), it would be interesting to see whether scavengers differ from predators. In the American kestrel, the contrast sensitivity of a non-stationary (reversed) pattern (vs. stationary pattern) is higher [67]. In humans, contrast sensitivity increases significantly with a higher speed of movement [86]. Both results suggest a better detection of moving versus stationary objects. Predatory raptors may be better at detect moving targets, such as active prey, as opposed to stationary targets, such as carcasses. In contrast, because scavengers forage almost exclusively on carrion, the ability to rapidly detect stationary objects (to overcome potential competition with predators that scavenge, or other scavenging species) may be more important in these raptors. However, other non-raptorial species that seek stationary items (e.g., Common starling *Sturnus vulgaris*, Japanese quail *Coturnix japonica*, and Rock dove *Columba livia*) have similar contrast sensitivity to predators [84]. Furthermore, it has been shown that the first raptors to arrive at a carcass are often not vultures, but eagles [87], which would seem to counter the hypothesis that vultures should have higher contrast sensitivity for stationary objects. Social information is also essential for scavengers [88] and vultures also use moving objects (conspecifics) to find their food. Social information significantly facilitates foraging success in vultures [89], potentially because detecting conspecifics (moving objects) is also easier for scavengers than detecting carcasses (stationary objects). In order to better understand if different foraging tactics reflect different contrast

sensitivity abilities, and if detection abilities by scavengers are improved by non-stationary objects, there is an urgent need to study scavenger species in detail.

*4.2. Temporal Resolution of Vision*

While spatial resolution has been studied in some raptors, including both predatory and scavenging species, the temporal resolution of vision (assessed as flicker fusion frequency, the frequency at which an intermittent light stimulus appears to be steady) has only been estimated in three predatory species, the Peregrine falcon *Falco peregrinus* (up to 125 Hz), the Saker falcon *Falco cherrug* (up to 102 Hz), and the Harris's hawk (up to 78 Hz) [90]. It has been argued that high temporal resolution should confer a selective advantage for fast-flying and manoeuvring species seeking for fast-moving prey [91]. While other species seeking stationary items, such as seeds, have high temporal resolution (e.g., up to 105 Hz in the Domestic chicken *Gallus domesticus* [92]), this seems to be confirmed in raptors, with the Peregrine falcon, the fastest animal in the world when diving on fast-moving prey [93], having the highest temporal resolution of vision among the three tested species [90]. Scavengers, especially vultures, are known to forage for carrion mainly using soaring flights [12], where rapid vision should not be necessary. However, high temporal resolution would benefit species with high spatial resolution (like large vultures) by reducing motion-induced blur [94]. Therefore, studying the temporal resolution of a broader range of raptor species and in particular some scavengers will be important to understand whether relatively high temporal resolution is common among diurnal raptors or restricted to species with predatory habits.

*4.3. Colour Vision*

Birds commonly use colours for discriminating object of interest, such as food, mates, or predators [95]. Most birds have a tetrachromatic vision system [96]. An important variation is found in the SWS1 cone pigment, which in birds can exist as either (1) a UV (ultraviolet) pigment or (2) a violet (V) pigment [97].

There is growing evidence that most raptors are unable to detect UV lights. Lind et al. (2013) measured the transmittance of the ocular media (cornea, lens, vitreous), which sets the limit of UV sensitivity, in four predatory raptors (the common buzzard *Buteo buteo*, the European sparrowhawks *Accipiter nisus*, the red kite *Milvus milvus*, and the Common kestrel *Falco tinnunculus*), and found UV cues are unlikely to provide a reliable visual signal to hunting raptors [98]. This is in contradiction with older behavioural studies, suggesting that vole scent marks are detectable in UV light by the Common kestrel [99,100]. Ödeen and Håstad (2013) published a study on the molecular biology of opsin genes in three other raptor species (the Turkey vulture, Pacific baza *Aviceda subcristata*, and Mississippi kite *Ictinia mississippiensis*) and found that these species have violet (but not UV) sensitive SWS1 cones [97]. Consequently, they suggested that Falconiformes and Accipitriformes cannot see UV. In 2016, Wu et al. (2016) studied the presence/absence of cone opsins in 11 additional diurnal raptor species (4 Falconiformes and 7 Accipitriformes) and found that SWS1 cone opsins were present in all species (without differentiating between violet-sensitive or UV-sensitive subtypes), except in two Accipitriformes: the Cinereous vulture *Aegypius monachus* and the Black winged kite *Elanus caeruleus* [101]. While further studies are needed to fully understand the role of UV light in raptor foraging, there is increasing evidence that most raptors cannot see UV, except maybe the Western marsh harrier *Circus aeruginosus* (Olsson, Mitkus, and Kelber, unpublished data).

Among the aforementioned species, only two scavengers have been considered (the Turkey vulture and the Cinereous vulture). Interestingly, the spectral sensitivity of these species, which are not considered to be members of the same taxonomic order, appears to differ, with the Turkey vulture (Cathartiformes) having tetrachromatic vision, but the Cinereous vulture (Accipitriformes) only having trichromatic vision [97,101]. Furthermore, the detailed spectral sensitivity of scavengers has never been studied contrary to predators (Common kestrel, Sparrowhawk *Accipiter nisus*, and Common buzzard *Buteo buteo* [98]).

Finally, the use of colour information appears to be important at a distance in raptors, with Harris's hawk having twice as good a chromatic spatial resolution compared with humans, while achromatic vision is similar [9]. The spatial resolution of a chromatic vision has only been estimated in two bird species, Harris's hawk [9] and the non-raptorial budgerigar *Melopsittacus undulatus* [102]. While spatial resolution of the chromatic vision is systematically lower than for the achromatic channel, understanding whether scavengers can discriminate colours at a distance would be essential to understand which cues are important to these birds for detecting carrion.

## 5. Conclusions

While raptors are often cited as having extraordinary sight, what I have highlighted here is that we actually have scarce knowledge on the visual sense in the vast majority of diurnal raptors, and especially in scavengers. Here, I reviewed what is currently known about the functional differences in the visual abilities of predators and scavengers. Predators have a visual system adapted to predation, characterised by the following: (1) large eyes with two foveas, one located centrally for the detection of prey at distance, and one located temporally to fixate the prey at the moment of capture and (2) a visual field with an enlarged binocular field to facilitate the guidance and positioning of the feet during prey capture, and a larger blind spot above the head to avoid sun-dazzling for species chasing preys on the ground. In contrast, scavengers have a visual system adapted to carrion eating and social foraging: (1) smaller eyes (compared with body mass) with only one central fovea to spot carrion at distance and (2) an enlarged field of view to facilitate the detection of carcasses and for the accurate use of social information. Despite these differences, it is important to remember that the visual systems of predators and scavengers are also very similar in a number of ways. For example, they both have the following: (1) a classical vertebrate optical system; (2) an identical design of the retinal layers; and (3) spatial resolution that appears to be adapted to flight altitude during foraging rather than diet per se, with some species having the highest visual acuity in the Animal Kingdom.

With raptors globally suffering from population declines [82], increasing our knowledge about their sensory ecology and behaviour could offer significant benefits to conservation programs [103]. In particular, scavengers, and especially Old-World vultures, are more threatened than any of the other groups [82], with over 80% of species declining. It has been shown that understanding their visual systems may help protect them, such as understanding the high collision rate with human devices [81]. Therefore, in light of our minimal knowledge on the sensory biology of scavenging raptors, there is urgency in studying their visual capacities.

**Supplementary Materials:** The following are available online at http://www.mdpi.com/1424-2818/12/10/400/s1, Table S1: Information about foraging tactics, corneal diameter, axial length and body mass of raptors.

**Funding:** I gratefully acknowledge support from the K. & A. Wallenberg Foundation, Stockholm (Ultimate Vision) and the Swedish Research Council (2016-03298_3).

**Acknowledgments:** I sincerely thank A. Kelber, M. Mitkus, and T. Lisney for their proof-reading and constructive comments on this review.

**Conflicts of Interest:** The author declares no conflict of interest.

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
