# Peer review of "Visual Adaptations in Predatory and Scavenging Diurnal Raptors"

_diversity, doi:10.3390/d12100400_

Round 1

Reviewer 1 Report

This is an interesting overview that describes and summarises the current understanding of visual adaptations in predatory and scavenging diurnal raptors. The author made an excellent review of the literature in this area and made comparisons between predatory and scavenging diurnal raptors.
The only comments I have about the manuscript are as follows:

1) No clear indication of the goals/hypotheses that the author would like to achieve/test in this work

2) Unfortunately, I have not found in the text a reference to part C of Figure 2

3) In my opinion, it would be good to discuss and further highlight the fact that vole urine is unlikely to provide a reliable visual signal to hunting raptors.
As suggested by earlier works, e.g.: Vitala et al (1995) and Honkavaara et al. (2002) and what seemed to be a well-established fact in ecology so far.

Author Response

Response to Reviewer 1 Comments

Point 1: This is an interesting overview that describes and summarises the current understanding of visual adaptations in predatory and scavenging diurnal raptors. The author made an excellent review of the literature in this area and made comparisons between predatory and scavenging diurnal raptors.

Response 1: I am delighted you enjoyed the manuscript. Thank you

Point 2: No clear indication of the goals/hypotheses that the author would like to achieve/test in this work

Response 2: You are right, this was not explicitly written in the introduction. I have now added a paragraph to explain the importance as well as the goals of such review (lines 77-88).

Point 3: Unfortunately, I have not found in the text a reference to part C of Figure 2

Response 3: Sorry this was a mistake. We have now included the reference of figure 2C in the text (lines 267, 271).

Point 4: In my opinion, it would be good to discuss and further highlight the fact that vole urine is unlikely to provide a reliable visual signal to hunting raptors.
As suggested by earlier works, e.g.: Vitala et al (1995) and Honkavaara et al. (2002) and what seemed to be a well-established fact in ecology so far.

Response 4: This is an interesting point of view. When I started to write the review, I included the reference of Viitala. However, after discussion with colleagues, we (and I) considered that it was better to not include this reference in the manuscript. Indeed, we believed that it was better to simple not mention it rather than refute it with the novel knowledge.

Nevertheless, I think it is good for readers to see that some works can be contradictory. I therefore included a sentence on Viitala work lines 339-341 (the added sentence is in italic below):

There are growing evidences that most raptors are unable to detect UV lights. Lind et al. (2013) measured the transmittance of the ocular media (cornea, lens, vitreous), which sets the limit of UV sensitivity, in four predatory raptors (the common buzzard Buteo buteo, the European sparrowhawks Accipiter nisus, the red kite Milvus milvus and the Common kestrel Falco tinnunculus), and found UV cues are unlikely to provide a reliable visual signal to hunting raptors [97]. This is in contradiction with older behavioural studies suggesting that vole scent marks are detectable in UV light by the Common kestrel [98, 99].

Reviewer 2 Report

Dear Author,

It is nice review, quite easy to read, but I will expect that you will clearly define the reason of such a review, what new you want to present to readers etc. Now the paper suits more to the book as a chapter, but not as a paper in the journal. It is interesting, some nice analyses are presented (Fig. 3) but could be better refined. Some improvements and comments I have added straigh to the text on pdf.       

Author Response

Response to Reviewer 2 Comments

Point 1: It is nice review, quite easy to read, but I will expect that you will clearly define the reason of such a review, what new you want to present to readers etc. Now the paper suits more to the book as a chapter, but not as a paper in the journal. It is interesting, some nice analyses are presented (Fig. 3) but could be better refined. Some improvements and comments I have added straight to the text on pdf.       

Response 1: I am sincerely grateful about the work you produced to review this manuscript. I have taken into account all your specific comments added in the PDF (and described below).

I have now clearly defined the reason of the review in the introduction, as well as the novelty of this one compared to previous reviews on raptor vision (lines 77-88).

I have also followed your comments about the figure 3 (see below).

However, I do not understand why this paper suits more to a book rather than a paper in the journal. This review is part of a special issue named as “Ecology of predation and scavenging and the interface”. I have been invited to write a paper for this special issue and I strongly believe that reviewing the visual adaptations of predatory and scavenging raptors suits perfectly to this issue. Maybe I did not understand your point.

Point 2: Probably the abstract could be changed after changes in the main text.

Response 2: I have followed the recommendation of both reviewers in the main text; however, I do not think the abstract has to be changed. In its actual form, the abstract clearly describes the purpose and the content of the paper.

Point 3: At this moment there is no clear aim of this review. Author should clearly inform readers what is the reason of such review, what new knowledge could be expected from gathering and summarising of all the data etc. Now the paper is nice to read, especially for readers that are not very familiar with the subject, but clear novelty, or at least a need of such review should be clearly presented in the introduction.   

Response 3: You are right, the goals/hypotheses of the review were not described in the introduction. I have now added a paragraph in the revised manuscript (lines 77-88). Because raptors are declining considerably worldwide, and because conservation biology deeply benefits from sensory ecology, it is important to highlight the visual adaptations of those species. Especially, among raptors, vultures are the most threatened group. Because almost all vultures are considered to be exclusive scavengers, they have different sensory demands compared to predators. Therefore, highlighting the similarities and the differences in the visual system of scavengers and predators may be essential in the perspective of raptor conservation.

Point 4: rather [4, 5, but see 6] 

Response 4: Done

Point 5: rather [7, 8, reviewed in 5], in this way it will be easier for read.

Response 5: Done

Point 6: rather (rollers, kingfishers and bee-eaters). Only (bee eaters), when the clad is illustrated by a kingfisher, not mentioned in the caption looks odd. 

Response 6: Thank you for pointing out this mistake. I have now modified it in the revised manuscript (line 56).

Point 7: Probably somewhere should be mentioned the source the author based on considering predators, opportunists and scavengers.

Also - opportunists were not clearly define here -as authors wrote above "While precise terminology is essential in science,.." he should clearly defined the group of opportunists.

Response 7: You are right. Raptors have been categorized as predators, scavengers and opportunists according to Wilman et al. (2014). When species were not present in Wilman et al. (2014), foraging specialization has been categorized according to Fergusson-Lees (2001). Specifically, all vultures were considered as scavengers, caracaras were all considered as opportunist based on diet and foraging strategies and species that scavenge at least 40% of their time were also considered as opportunists as well. Species that scavenge occasionally (<20% of their time) were considered as predators. Because this was not specified in the review, I added this information in the revised manuscript.  I added this information in the revised version of the manuscript (lines 80-85).

Point 8: Also it could be very useful if author will estimate number of predators, opportunists and scavengers among studied diurnal raptors. 

Response 8: This is a very good point. I added this information in the revised manuscript concerning the corneal diameter (see Figure 3). Moreover, throughout the manuscript, I specified to which categories the species belong.

Point 9: here - in the caption, or above the species names in the figure (better option) the foraging tactic should be presented. Now the tactic is mentioned, but species are not assigned.

Response 9: This is a very good point. I have now added the foraging tactics in the Figure 2.

Point 10: I like this paper and figures, but in my opinion too much is presented in the caption. Results of this fig probably could be better elaborated/discussed in the text.  

Response 10: Thank you for this comment. I questioned myself a lot about the caption of this figure and the analyzes included within this caption. I have made the choice to put it in the caption for different reasons.

First, this is the only statistical analyze included in the review. I strongly believe that incorporating this statistical analyze in the main text would reduce considerably the readability of the manuscript. I do not think that presenting the methodology and the statistical results in the main text would be beneficial.

Second, in this specific context, I think it is good for the reader to have directly access to the significant differences within the caption. We have made this choice in previous review and in this case, I believe this is the best solution.

Therefore, based on this reflection, I have made the choice to keep the caption as it was in the first version of the manuscript. I may have wrong and if you (and the editors) consider that this is essential to include it in the main text, I can modify it.

Point 11: what is presented here?- mean with SE/SD/25-75% and ranges? Dots represent species? It should be clear. Also it could be useful when number of analysed species will be presented for each foraging tactic (e.g. in brackets)

Response 11: I have added all necessary information in the revised version of the manuscript (see Figure 3).

Point 12: rather in raptors - if this was calculated for data from panel A. In the panel trend line will be useful. 

Response 12: Sorry for this mistake. I have now corrected it in the revised version of the manuscript (line 131).

Point 13: I'm not sure whether Chi square test is proper here, as for such presented data I will expect e.g. Kolmorov-Smirnov test, but maybe phylogenetic relations were also included here, or it is specific result from the R environment.   

Response 13: Sorry if it was not clearly described. I did not perform a chi square test but I compared two phylogenetic models (one including the foraging tactic and one without the foraging tactic as fixed effect) using a likelihood ratio test, based on Chi-squared distribution (lrtest function from lmtest package). This is why the output of the model looks like a chi square test. I have added more information in the revised version of the manuscript (lines 137-139).

Point 14: From

Response 14: Done

Point 15: but the diet is not studied here, but rather foraging mode

Response 15: You are right. I have corrected it in the revised manuscript (line 218).

Point 16: So what is new here? maybe at least mean could be calculated? 

Response 16: As added in this revised version of the manuscript, the novelty of this review is to highlight the differences and similarities of scavengers and predators. I have explained it in the revised version of the manuscript lines 77-88.

I do not agree that mean is a relevant measure. As highlighted in the manuscript, raptors differ significantly in terms of ecology and phylogeny. Therefore, they differ considerably in their visual demands. Presenting the average spatial resolution would not be relevant here as this would not mean anything. Also, because spatial resolution has been estimated using different methods that may not be comparable (see lines 188-190), there is no reason to calculate the mean. By contrast, this would confuse the readers. Therefore, I have not added the average spatial resolution of raptors in the manuscript.

Point 17: probably it could be useful to mention for such results papers dealing with vole urine UV‐reflectance and foraging in some raptors. 

Response 17: The first reviewer also suggested to add the papers dealing with vole urine UV-reflectance and foraging. Please see my response to the first reviewer below:

This is an interesting point of view. When I started to write the review, I included the reference of Viitala. However, after discussion with colleagues, we (and I) considered that it was better to not include this reference in the manuscript. Indeed, we believed that it was better to simple not mention it rather than refute it with the novel knowledge.

Nevertheless, I think it is good for readers to see that some works can be contradictory. I therefore included a sentence on Vitala work lines 339-341 (the added sentence is in italic below):

There are growing evidences that most raptors are unable to detect UV lights. Lind et al. (2013) measured the transmittance of the ocular media (cornea, lens, vitreous), which sets the limit of UV sensitivity, in four predatory raptors (the common buzzard Buteo buteo, the European sparrowhawks Accipiter nisus, the red kite Milvus milvus and the Common kestrel Falco tinnunculus), and found UV cues are unlikely to provide a reliable visual signal to hunting raptors [97]. This is in contradiction with older behavioural studies suggesting that vole scent marks are detectable in UV light by the Common kestrel [98, 99].

Round 2

Reviewer 2 Report

Authors nicely responded to my comments, and corrected the manuscript. I was not aware that the paper was invited to the special issue, so maybe some of my my previous comments were too strong. But I hope that were useful for authors, and now the paper is easier to read, the aim of the review is clear, figures corrected etc, so I'm satisfied with authors work.